# Predicting Response to Neuropsychological Intervention in Developmental Dyslexia: A Retrospective Study

**DOI:** 10.3390/brainsci14080775

**Published:** 2024-07-31

**Authors:** Maria Luisa Lorusso, Francesca Borasio, Simona Travellini, Massimo Molteni

**Affiliations:** 1Unit of Child Psychopathology, Scientific Institute IRCSS E. Medea, 23842 Bosisio Parini, Italy; francesca.borasio@lanostrafamiglia.it (F.B.); massimo.molteni@lanostrafamiglia.it (M.M.); 2Department of Humanistic Studies (DISTUM), University of Urbino “Carlo Bo”, 61029 Urbino, Italy; simona.travellini@uniurb.it

**Keywords:** response to intervention, predictor, reading, writing, VHSS, RAN, phonological awareness, visual search

## Abstract

Identifying the patients who are likely to be non-responders to a certain treatment may allow clinicians to provide alternative strategies and avoid frustration and unrealistic expectations for the patients and their families. A retrospective study on 145 children treated with visual hemisphere-specific stimulation examined the specific profiles (reading, writing, metaphonology, memory, callosal functions) of non-responders, and identified predictors of response to intervention (reading, reading and writing) through linear regression models. The effects of additional variables such as rapid automatized naming (RAN) and Visual Search were investigated in a subsample of 48 participants. Subgroups related to gender and dyslexia subtype were considered in the analyses. The results highlight an Intervention Differential Effect (IDE) not depending on regression to the mean and mathematical coupling effects. The characteristics of non-responders for reading seem to correspond children with mild reading and severe writing impairments; non-responders for reading and writing are those with impaired callosal transfer. Predictors of overall response to intervention were pre-test reading and writing scores; phoneme blending, accuracy in visual search and speed in rapid automatized naming contributed to explaining response variance. Specific predictors for female vs. male participants and dyslexia subtypes were identified.

## 1. Introduction

Developmental dyslexia (DD) is one of the most frequent learning disabilities, and it is diagnosed in children who fail to develop normal reading skills in spite of normal intelligence, in the absence of sensory or neurological damage [1,2]. Over the past decades, there has been an increasing interest in understanding factors that underlie the development of reading skills and response to reading intervention.

Recently, a number of studies have supported multiple-deficit theories of DD, assuming that no single deficit is sufficient to explain all children’s reading profiles [3,4,5]. The multiple-deficit view of DD leads to creating and providing tailored interventions, focused on each individual’s needs [6,7,8].

Although there are several effective interventions for reading, for a minority of children the response to the treatment appears poor and the difficulties remain severe. For instance, in a study adopting combined elements from the reading and the phonology approaches [9], the responsiveness of the children varied, and about a quarter of the group could be defined as treatment “non-responders”. The term referred to children who had not shown a gain in standard score on the reading test despite improving their letter–sound knowledge and phoneme awareness. In that case, children were typically those with more severe phonological impairments, poor vocabulary skills, and problems in attention control. Phonological awareness deficits appear to be present in the majority of dyslexic children who are non-responders to early literacy intervention [10].

Other studies on response to intervention have shown similar findings: poor rapid automatized naming, poor phoneme awareness, poor memory, poor oral language/vocabulary, and problem behaviors were the predictors of the outcomes of the treatment [11,12,13]. Cognitive factors, instead, appeared not to be strong predictors of intervention outcomes after accounting for baseline measures of reading [14,15].

In a recent meta-analysis, phonological awareness and rapid automatized naming were identified as very good predictors of reading outcomes, accounting for approximately 20% to 25% of the variance in response to intervention in reading [15]. More recent studies [16,17,18] confirmed the role of RAN, verbal WM and phonological awareness in predicting response to intervention for reading difficulties. Understanding the role of specific cognitive skills may help improve the detection of risk factors and prediction of individual response to interventions.

Remediation programs very often target phonological deficits, known to be among the strongest predictors of word reading accuracy [19,20]. However, in the framework of multi-factor models of dyslexia [3] it has become more and more evident that targeting also other mechanisms involved in the reading process may be justified at the theoretical level and advantageous for the purposes of clinical improvement [6,7]. Spatially and temporally dependent processes, such as visual-attention, rapid naming and word recognition, as well as eye movement, are indeed predictors of reading fluency [21,22]. Among the intervention programs addressing a wide range of different reading-related processes, visual hemisphere-specific stimulation (VHSS), according to Bakker’s Balance Model [23,24] and recent revisits [25,26,27], even if based on an outdated view of reading and dyslexia in neuropsychological and neurofunctional terms, has shown to be particularly appealing for the thorough description of a very systematic methodology to address the various stages of reading acquisition, initially resting on more visual-based decoding and progressively relying on more language-based anticipation strategies for efficient reading. This approach thus aims to activate both lexical and sub-lexical processes that help consolidate orthographic representations, but also devotes a large amount of work to reading strategies at a metacognitive level, and has proven to be one of the most effective programs for DD in the Italian background [28]. VHSS treatment supports the training of more perceptually-oriented, sublexical analysis of words with direct stimulation (via tachistoscopic presentation to the left visual field) of the right hemisphere, whereas more language-based anticipation strategies (addressing the lexical, phonological, morphological, syntactic and semantic characteristics of the stimuli to be read) are strengthened by predominant stimulation of the left hemisphere (via tachistoscopic presentation to the right visual field). Even if the exact neurofunctional mechanisms activated by this type of stimulation are not completely clarified and the hypothesis of nonspecific effects has been put forward [29], it has been experimentally shown that visual hemisphere-specific stimulation (excluding hemisphere-alluding stimulation and tactile stimulation) does produce an advantage over non-specific or reversed stimulation [26]. Moreover, neuroimaging evidence showing different contributions from each of the two hemispheres to reading during reading acquisition, the right hemisphere being predominantly activated in early stages whereas the left hemisphere increases its involvement in later stages [30,31,32], has lent indirect support to the basic principles of the program. To the aim of the VHSS intervention, three types of dyslexic readers are distinguished: P-types are characterized by relatively slow but accurate reading, L-types by relatively fast but inaccurate reading, and M- types (mixed) by both slow and inaccurate reading [23,26,33,34]. This subtyping approach mirrors (in spite of some differences) more recent typologies proposed for instance by [35,36,37,38] for readers of Dutch, Hebrew, and Arabic.

Several studies showed the effectiveness of VHSS in improving reading skills, promoting stable enhancement in both reading fluency and accuracy [22,24,26,28,32,39,40]. In particular, lateral visual presentation is especially beneficial to reading accuracy in L- and P-types, while central stimulation involving both hemispheres simultaneously seems to be most effective in enhancing spelling abilities [26]. VHSS has been shown to improve phonemic awareness, spatial attention, automatization, auditory short-term and working memory and memory retrieval processes [26,33], all of which have, in turn, been claimed to be predictors of reading ability.

Evidence shows that the two strongest predictors of learning to read in the early stages are phoneme awareness and automaticity of letter–sound knowledge [41,42]. Other predictors of reading include auditory rote memory, working memory, visual memory, attention and vocabulary [43,44,45]. Studies conducted in transparent writing systems, such as Spanish, German or Italian, have confirmed the predictive role of phonological awareness on dyslexia [46], even if in some cases its impact on reading difficulties decreases after the first years of schooling, becoming a weaker predictor of reading acquisition than in opaque orthographies [47,48]. Also, automaticity of lexical access as measured by RAN (sometimes considered to be a different expression of phonemic awareness) has been shown to predict reading acquisition in both Western and Eastern orthographic systems [49,50]. Regarding predictors different from phonemic awareness, an Italian study showed that performance on visual attention tasks in preschoolers predicted reading ability two years later [22].

Predicting reading ability and predicting response to intervention, however, may represent two very different mechanisms, especially when intervention is not a school-based training but a clinical treatment based on neuropsychological models. As highlighted also by Stuebing et al. [15], predicting responses to intervention can have a more theoretical (and in this case, among other procedures, adjusting for initial levels may be reasonable) or a more clinically oriented goal (identifying patients who are at risk of not responding positively to intervention, in order to offer different, personalized solutions and strategies, prevent useless efforts from both the patient’s and the service’s side, as well as unrealistic expectations and likely disappointment by the patients and their families). The present study moves in the latter perspective, exploring the specific markers of non-responders as well as the predictors of improvement in reading and spelling skills after a neuropsychological treatment based on VHSS, in children with a diagnosis of DD and in a transparent orthography. The study is retrospective as data were collected over six years at IRCCS ‘Eugenio Medea’. Although inclusion criteria were the same over time, as well as tests of reading and writing ability, memory and phonemic awareness, some additional measures were collected only during certain periods, linked to specific research protocols. These additional measures include callosal transfer of tactile information, RAN (rapid automatized naming) and visual search.

## 2. Materials and Methods

### 2.1. Participants

To the aim of the present retrospective study, data from 145 children treated with VHSS were collected. The main group of 145 children were aged between 7 and 15 years (M = 9.60; SD = 1.73; female = 53, male = 92). The effects of additional variables such as rapid automatized naming (RAN) and visual search were investigated in a subsample of 48 participants. Table 1 shows descriptive statistics at pre- and post-test assessment.

Children had to fulfil the following inclusion criteria: (a) having been diagnosed with DD (ICD-10 code: F81.0) on the basis of standard inclusion and exclusion criteria [2] (at least one z-score concerning reading/writing speed and/or accuracy below −2, normal or corrected-to-normal sight and hearing, absence of neurological pathologies, regular schooling); (b) absence of comorbidity with other neuropsychiatric or psychopathological conditions (whereas comorbidity with other learning disorders and/or ADHD were allowed); (c) not having been involved in other reading intervention programs. All children were native speakers of Italian. No information about SES was available in the database.

Participants were recruited among patients of the child neuropsychiatry unit of Scientific Institute IRCCS ‘Eugenio Medea’ in Bosisio Parini, Northern Italy. Written parental informed consent was obtained before the beginning of the treatment; research consent was not provided given the retrospective characteristic of this study. The study was approved by the Local Ethics Committee in accordance with the Declaration of Helsinki.

### 2.2. Neuropsychological Assessment

All participants were tested before and after treatment. The results of the tests are expressed as z-scores according to age norms, with the exception of the raw scores of the phonemic awareness and callosal transfer tasks, for which no age-norms are available. The following tests were administered in the pre- and post-test sessions, with the exception of the cognitive measures which were adopted only in the pre-test session to assess inclusion criteria related to normal intelligence.

Cognitive measures: The Wechsler Intelligence Scale for Children, Revised and Third edition version (WISC-R and WISC-III) [51,52] was adopted to assess inclusion criteria related to normal intelligence.

Text reading: “Prove di rapidità e correttezza nella lettura del gruppo MT” (“Test for speed and accuracy in reading, developed by the MT group” [53,54] was used to assess reading abilities for meaningful material. The test provides separate scores for speed and accuracy in reading age-related texts (grades 1–8 of primary school). This task was used for classification of dyslexia according to Bakker’s model (P-, L-, or M-type), based on error type and reading speed.

Single word/nonword reading: “DDE-2: Batteria per la Valutazione della Dislessia e Disortografia Evolutiva-2” (Assessment battery for Developmental Reading and Spelling Disorders-2) [55] was used to assess speed and accuracy (expressed in number of errors) in reading words (96) and nonwords (48). It provides grade norms from the second to the last grade of junior high school.

Spelling tests: “DDE-2: Batteria per la Valutazione della Dislessia e Disortografia Evolutiva-2” (Assessment battery for Developmental Reading and Spelling Disorders-2) [55] was adopted to assess accuracy (expressed in number of errors) in writing words (48), nonwords (24), and sentences (12).

Phonemic awareness [56] was assessed through two subtests. The test of phonemic elision assesses the ability to recognize and isolate the phonemic constituents of 20 words: the child is asked to delete the first two phonemes in the word read by the examiner and to report the resulting nonword. The phoneme blending task assesses the capacity to derive a phonemic pattern from distinct phonemic units: the examiner presents each of 20 words letter by letter and the child is requested to identify and report the resulting word. For both tasks, raw scores referring to the total number of errors were recorded and used.

Rapid automatized naming: “Denominazione Rapida” (rapid automatized naming test—RAN) [57] was used to assess the naming speed for familiar stimuli (figures, colors, digits). Two matrices (10 rows of 5 stimuli each) of each type of stimulus were presented and the child was asked to sequentially name each visual stimulus of the matrix as quickly and as accurately as possible. Two raw scores were recorded: speed, expressed in seconds, and accuracy, expressed in number of naming errors. Z-scores based on grade norms were calculated.

Visual search: “Ricerca visiva” (visual search—VS) [57] was used to assess speed and accuracy in visual search for familiar stimuli (figures, digits). Two matrices (10 rows of 5 stimuli each) for each kind of stimulus were presented and the child was requested to cancel one of the stimuli (the number “7” and the figure “star”) presented in the matrix as quickly and as accurately as possible. Two raw scores were recorded: speed, expressed in seconds, and accuracy, expressed in number of cancellation errors. Z-scores based on grade norms were calculated.

Callosal transfer: The Tactile Finger Localization Test (TFLT), derived from the test developed by Volpe et al. [58], was used to evaluate the ability to localize tactile stimuli on the proximal or distal phalanx of the fingers of one hand and to transfer that information to the opposite hand intermanually. In the present task the child was blindfolded and sat with their hands flat on the table, palms downwards. The experimenter, using the rubber end of a pencil, lightly touched one of 8 points of the fingers in random succession (proximal or distal volar surface of fingers 2–5; 3 stimulations per point, 24 random trials per hand). Immediately following stimulation, the child was asked to touch each stimulated point using the thumb of the same hand (intramanual task), or the same point in the contralateral hand using the thumb of that hand (intermanual task). All participants performed the intramanual trials first and intermanual trials afterwards. The number of errors in the intramanual and intermanual tasks were calculated, for the left and the right stimulated hand separately.

Verbal memory: “Test di Memoria e Apprendimento” (TEMA; Italian adaptation of TOMAL, Test of Memory and Learning) [59] was used to assess short-term memory and working memory, supra-span verbal learning, and long-term memory. Verbal short-term memory was assessed by means of immediate serial recall (span) of letters (Forward Letter Span). Working memory was evaluated by backward recall of letters (Backward Letter Span). The number of letters correctly recalled in the correct sequence was recorded. Supra-span verbal learning was assessed by means of a free-recall paradigm, in which the child is requested to learn a list of 12 words, in a series of maximum eight trials. Scores are the total number of correctly recalled words. Age norms are provided.

### 2.3. Treatment

#### VHSS Training

The children received visual hemisphere-specific stimulation (VHSS), based on a revisit of Bakker’s ‘Balance model’ [23,24], and according to the structured program of Lorusso and colleagues [26,33]. Each child was classified as P-, L-, or M-type dyslexic reader based on the persistent over-reliance on specific reading strategies, on reading speed and on the pattern of reading errors. Children were subdivided into three types of readers:-P-types (decoding strategies based on accurate perceptual analysis mainly supported by the RH, resulting in slow but relatively accurate reading) if reading speed is at least 1 SD below age mean and the proportion of time-consuming errors over total errors is ≥60%;-L-types (anticipation strategies based on linguistic abilities and mainly supported by the LH, resulting in relatively fast but inaccurate reading) if reading speed is no more than 1 SD below age mean and the proportion of substantive errors over total errors is ≥60%;-M-types (who strive to use both kinds of strategies but do so inefficiently, resulting in both slow and inaccurate reading) in all other cases.

VHSS aims at increasing the contribution of the less involved hemisphere to the reading process by manipulating sensory stimulation, stimulus characteristics and tasks. The tachistoscopic presentation of words to a visual hemifield selectively stimulates either the RH (additionally requesting RH-specific perceptual analysis using visually complex materials and/or error detection and correction tasks) or the LH (additionally requesting LH-specific linguistic anticipation using linguistically inter-related materials and/or anticipation/completion tasks). A program called ‘FlashWord’ [60] was used, in which the word is flashed on the PC screen (presented for less than 350 ms) in the target visual field only if the child clicks on the mouse at the exact moment a dot is crossing a central target (fixation point). Ad hoc created lists of stimuli and tasks stimulate linguistic anticipation based on semantic categorization, and/or on the morphosyntactic structure of the presented stimuli vs. decoding strategies based on the perceptual characteristics of the presented stimuli.

VHSS treatment was carried out in individual sessions twice a week, lasting 45 min each, over a 4-month period, and it was discontinued after exactly 32 actual sessions. The sessions took place at the outpatient clinic of the Scientific Institute IRCCS ‘Eugenio Medea’.

### 2.4. Data Analysis

Data were analyzed with SPSS software v.26, according to the following steps.

To the aim of the present study, mean response to intervention was calculated as the average of observed changes (post-test scores minus pre-test scores) in all measures. An average of all difference z-scores including speed and accuracy scores for all the administered tests (text, word and nonword reading) was calculated and labelled “mean response for reading”. “Mean response for reading and writing” was calculated as the average of the previous score and of the mean of all spelling and writing z-scores (accuracy scores for word, nonword and sentence writing to dictation). The rationale behind the choice of averaging several variables and parameters is that children with reading disorders may present with very different profiles, and show impairments on different abilities (e.g., some children are more affected in speed and others more in accuracy; similarly, some children may have larger impairments with nonwords, and others may find more difficulties with lexical material such as words and texts). Treatment will thus be focused on different parameters and skills for each of these cases, and success of treatment should be expressed as the improvement of the targeted ability, possibly accompanied by minor improvements in other abilities, and not counterbalanced by worsening in other skills (exactly what will be captured by the averaged scores). This approach is especially meaningful considering that the intervention model used in the present study is actually based on the existence of different subtypes of dyslexia and of specific intervention strategies tailored for each of the subtypes [23,24].

An a priori criterion was then applied based on the two response scores: the children whose improvement was less than 0.25 z-scores were labelled as non-responders (for either reading or reading-and-writing), while the children showing greater improvement were considered responders. The criterion was based on the observation that average improvement in performance was about 1 z-score with standard deviation being 0.93, thus a fourth of the average gain was considered to be an appropriate cut-off to indicate an irrelevant response. To confirm the validity of such cut-off, reading speed as measured in syllables per second was taken as an example. According to a study on the Italian population [61], untreated children with dyslexia increase their reading speed by about 0.3 syllables per second per grade, which could be estimated as 0.1 in four months (the duration of the treatment in our case). In our sample, the mean speed gain in (a) responders for reading was 0.37, compared to the mean speed gain in non-responders of 0.20, and improvements in (b) responders for reading and writing was 0.39, compared to mean speed gain in non-responders of 0.13, similar to what was reported by Tressoldi and colleagues [62]. This confirms that the subdivision identified children whose average speed gain was above the gain observed in untreated children in four months (responders) and children whose average speed gains were not clearly superior to those observed in untreated children (0.138 would be the minimum difference score, reaching a significant difference in a one-tailed t-test with respect to the improvement of untreated children).

Pre-test scores between the two groups of responders and non-responders were compared in order to identify possible markers of non-responders’ profiles. To reduce the number of variables and obtain more reliable scores based on multiple sources, nine global scores (in z-scores, except for phonemic awareness and callosal transfer) were computed for pre- and post-tests separately: a. Global accuracy score (the average of accuracy scores in text, word, and nonword reading), b. Global speed score (the average of speed scores in text, word, and nonword reading), c. Global writing accuracy score (the average of accuracy in word, nonword, and sentence writing from dictation), d. Global phonemic awareness error score (the sum of phonemic blending and phonemic elision scores), e. Global memory score (the average of scores for letter span forward and backward, and words free recall), f. Global RAN accuracy score (the average of accuracy scores for figures, colors, digits), g. Global RAN speed score (the average of speed scores for figures, colors, digits), h. Global visual search accuracy score (the average of accuracy scores for figures and digits), and i. Global visual search speed score (the average of speed scores for figures and digits), j. Global intramanual tactile errors (the average of errors for the left and the right hand), and k. Global intermanual tactile errors (the average of errors following stimulation of the left and the right hand). Subsequently, difference scores between the post-test and the pre-test were calculated, indicating improvement or worsening of performance. When significant or close-to-significant (i.e., *p* < 0.10, considering that a large effect of a single subcomponent could have been masked by small or null effects in other subcomponents) differences emerged between the two groups of children (responders and non-responders) at the pre-test on a global score; the scores of the components involved in it were analyzed with a multivariate analysis of variance to better characterize the difference (overall multivariate effect and single effects, with alpha = 0.05, two-tailed). If a significant multivariate effect was present, no correction was applied to the component single effects, otherwise Bonferroni correction was applied, depending on the number of variables composing the global score.

After group comparison, Pearson’s correlation coefficients were computed to explore the associations in the whole sample between mean response in reading and writing and initial values of reading and reading-related variables. Correlations were computed in the entire group of participants, including responders and non-responders. Since the pattern of significant correlations served as a basis to identify the variables to be included in regression models, it was decided to compute correlations on both the global scores and the atomic component variables. Atomic scores were used as substitutes for global scores whenever the analyses showed that large discrepancies existed among component scores and that only one or two of the atomic variables was a good candidate for a predictive model. In all other cases, global scores (more stable and more reliable than component scores) [62] were preferred to atomic scores for the regression models. In order to avoid spurious effects due to regression to the mean, which could inflate the correlation between initial values and observed improvement [63], all significant correlations involving reading and writing (i.e., the criteria used for diagnosis) were further checked applying Oldham’s method [64]. Only variables that passed the test (with alpha = 0.05), thus ensuring that observed associations did not merely reflect regression to the mean, were further considered for regression models.

As a final step, all significant associations (*p* < 0.1) between improvement in reading/writing and pre-test measures were used to construct multiple linear regression models (with forward stepwise selection of variables, and insertion cut-off = 0.05) to identify the best predictors of reading and writing improvements.

## 3. Results

### 3.1. Identification and Comparison of Responders and Non-Responders for Reading

To the aim of the study, children were divided into two groups according to their response to intervention, and performances at pre-test were compared. Based on the mean response in reading, 128 children were classified as responders while the remaining 17 (11.7%) were labelled as non-responders (less than 0.25 average z-score improvement). The analysis of variance on IQ and age showed no significant differences between the two groups (all *ps* > 0.288). As to pre-test global scores, a multivariate analysis of variance showed a significant difference between responders and non-responders, F (7,137) = 3.087, *p* = 0.005, partial eta square = 0.136. Of the seven included global scores (reading accuracy, reading speed, writing accuracy, phonemic awareness, memory and callosal transfer), a significant difference emerged for Global writing accuracy (*p* = 0.029, partial eta square = 0.033), while Global reading accuracy and Global phonemic awareness almost reached significance (*p* = 0.081, partial eta square = 0.023 and *p* = 0.071, partial eta square = 0.021, respectively). Pre-treatment scores and group comparisons are presented in Table 2.

As planned, the component scores of significant or close-to-significant variables were further analyzed with multivariate general linear models. Further analyses on global score components showed a significant multivariate effect of Global writing accuracy (*p* = 0.013, partial eta square = 0.080), and single effects of nonword (*p* = 0.003, partial eta square = 0.063) and sentence writing (*p* = 0.025, partial eta square = 0.038). Moreover, significant multivariate effects emerged for Global reading accuracy (*p* = 0.037, partial eta square = 0.058) and a single significant effect of nonword reading accuracy (*p* = 0.016, partial eta square = 0.040). No significant differences emerged for phonemic awareness subcomponents.

### 3.2. Associations between Reading Improvements and Pre-Test Measures

The relationships between mean response in reading and all the pre-test measures were explored in the entire group of participants (n = 145). Pearson’s correlation indexes’ global scores are presented in Table 3.

Improvements in reading were significantly, negatively, associated with Global reading accuracy, Global reading speed, and Global intermanual tactile transfer errors, while no significant associations emerged with Global writing accuracy, Global phonological awareness, Global memory score or Global intramanual tactile errors. In planned post hoc analyses of subcomponents, significant associations of mean response in reading emerged specifically with text, word and nonword reading speed (r ≤ −0.350, *p* ≤ 0.001) and accuracy (r ≤ −0.179, *p* ≤ 0.031), while all the other subcomponents of global scores did not reach significance (all *p*s ≥ 0.107). Significant correlations with Global reading scores were computed again following Oldham’s procedure, controlling for possible effects of regression to the mean: the results confirmed previous analyses with respect to Global reading speed (significant correlations with Global reading speed averaged between pre-and post-test, *p* < 0.05), suggesting that the correlations with Global reading speed scores were robust and not specific to single subcomponents. By contrast, correlations with global and atomic reading accuracy scores were not confirmed (all r in absolute value ≤ 0.062, all *p* ≥ 0.459), suggesting that regression to the mean could be inflating correlations at pre-test.

#### 3.2.1. Regression Analyses

Based on correlations in the whole group of children (n = 145), a regression analysis was conducted where the mean response for reading was used as a dependent variable, while Global reading speed and Global intermanual tactile transfer errors were used as predictors. Only Global reading speed entered the model (F = 23.91, *p* < 0.001) and accounted for 14.3% of total variance. Since the amount of variance explained by this model was rather limited, we decided to explore predictive models taking into account a larger number of variables, even if in a reduced sample of participants.

#### 3.2.2. Correlations and Regression Models in Children with Extended Assessment (n = 48)

The effects of additional variables such as RAN and Visual Search were investigated in a subsample of 48 children. In this subgroup, mean response for reading was significantly associated with Global reading speed and accuracy, Global writing accuracy, Global memory, and Global RAN speed. Considering subcomponents of significant global scores, mean response for reading was associated with text and word reading accuracy (r ≤ −0.460, *p* ≤ 0.001), with text, word and nonword reading speed (r ≤ −0.373, *p* ≤ 0.009), with word writing accuracy (r = −0.352, *p* = 0.014), phonemic blending (r = 0.288, *p* = 0.047), and letter span forward (r = −0.328, *p* = 0.023). Moreover, reading improvements were associated (*p* < 0.10) with Global intermanual tactile errors, Global Visual search accuracy, and nonword writing accuracy (r = −0.253, *p* = 0.082). Pearson’s correlations of Global measures (two tailed) are presented in Table 4.

Again, all significant correlations with reading/writing measures were checked with Oldham’s method: they were all confirmed (r in absolute value ≥ 0.310, *p* ≤ 0.032), with the exception of Global writing accuracy as well as its components. This resulted in writing measures being excluded from subsequent regression models, as they were likely to reflect spurious effects related to regression to the mean.

Based on correlations, a regression analysis was conducted where the mean response for reading was used as a dependent variable, and Global reading speed and accuracy, phonemic blending, Global intermanual tactile errors, Global RAN speed, Global visual search accuracy and letter span forward were used as predictors. A regression model with Global reading speed and Global reading accuracy as predictors was found to be significant (F = 14.31, *p* < 0.001) (Figure 1), and predicted 38.9% of variance (27.1% accounted for by reading speed and 11.7% accounted for by reading accuracy, both with negative beta values).

The quite different pattern of results found in the subgroup of 48 children as compared to the original group of 145 participants was rather surprising, especially since the subgroup simply collected participants who had been referred to the institute in a certain period of time and no selection criteria had been applied. Since the composition of the larger and smaller group was not perfectly homogeneous both in terms of gender (63.4% males in the large group vs. 47.9% males in the subgroup) and—to a lesser extent—in terms of subtypes (9.7% L, 30.3% P and 60% M in the large group vs. 8.3% L, 27.1% P and 64.6% M in the subgroup), we hypothesized that different variables could be predictive in different subgroups (while their effect in the whole group would be reduced due to heterogeneity and contrasting trends). Therefore, further correlation and regression analyses were computed in different subgroups of children, in particular (a) male and female, and (b) children classified as P-, L- or M-type dyslexic readers. In order to avoid excessively reducing the size of the subgroups, all analyses were computed excluding RAN and visual search variables that were available only for a subset of children. Lastly, correlation and regression analyses were computed for the (c) M-type dyslexic children (representing almost 65% of the sample and thus allowing for a sufficiently large number of participants with full assessment) belonging to the subgroup of 48 children. 

#### 3.2.3. Correlations and Regression Models in Male and Female Children

Pearson’s correlation analyses (two-tailed) were carried out separately in the male (n = 92) and the female group (n = 53). In the male group, mean response for reading was significantly associated with Global reading speed (r = −0.336, *p* < 0.001) and Global reading accuracy (r = −0.411, *p* < 0.001), in all their components. Moreover, nonword writing accuracy and left intramanual tactile errors almost reached significance (r = 0.207, *p* = 0.059 and r = −0.173, *p* = 0.099 respectively). In the female group, instead, significant associations emerged with Global reading speed (all components), nonword writing accuracy (r = −0.240, *p* = 0.090) and Global phonemic awareness (r = −0.237, *p* = 0.087). The check for spurious effects following Oldham’s method showed that correlations with accuracy scores for males could represent an effect of regression to the mean (r ≤ 0.064, *p* ≥ 0.547), whereas all effects related to reading speed were confirmed (r ≤ −0.314, *p* ≤ 0.002). Also, the correlation with nonword writing accuracy was confirmed (r = 0.315, *p* = 0.004). In the group of female children, no variable survived Oldham’s test (r ≥ −0.173, *p* ≥ 0.215), suggesting that correlations with pre-test Global reading speed and nonword writing accuracy were inflated by regression to the mean. Regression analyses were thus computed in the male group, with the mean response for reading as a dependent variable and Global reading speed, nonword writing accuracy and left intramanual tactile errors as predictors. The model with Global reading speed and nonword writing accuracy was significant (F = 9.03, *p* < 0.001) and accounted for 18.2.% of variance (14% Global reading speed, 4.2% nonword writing accuracy) (Figure 2). Global phonemic awareness did not enter the regression model computed in the female group.

#### 3.2.4. Correlations and Regression Models in Different Subtypes of Dyslexia

Based on Bakker’s Balance Model, participants were classified as L- (n = 14), P- (n = 44) and M-type (n = 87) dyslexic readers. Pearson’s correlation analyses of Global scores (two-tailed) are reported in Table 5.

In L-type dyslexic (fast and inaccurate) readers, mean response for reading was associated with pre-test accuracy in Global reading, essentially depending on text reading (r = −0.764, *p* = 0.001). In the P-type (accurate but slow) group, significant correlations emerged with Global reading speed in all its components of text, word and nonword reading speed (r ≤ −0.372, *p* ≤ 0.013), and with word reading accuracy (r = −0.317, *p* = 0.036). Finally, in the M-type group (slow and inaccurate), mean response for reading was associated with Global reading speed and all its components (text, words and nonwords, r ≤ −0.277, *p* ≤ 0.008), with Global reading accuracy (words and nonwords, r = −0.219, *p* = 0.042), and with phonemic blending (r = −0.178, *p* = 0.099). Checks for spurious effects with Oldham’s method confirmed validity for the correlation with reading accuracy in L-types (especially with text reading accuracy: r = −0.528, *p* = 0.052, whereas other components were not fully significant) and with reading speed in P-types (for all components: r ≤ −0.339, *p* ≤ 0.024), but not for word reading accuracy (r = −0.022, *p* = 0.887). For M-types, correlations with reading speed passed Oldham’s test under a one-directional hypothesis (r ≤ −183, *p* ≤ 0.089)—which could be acceptable considering the characteristics of M-types leading to consistent a priori hypotheses—but not those with reading accuracy (r ≤ 0.117, *p* ≥ 0.279).

Regression analyses were computed in the three different subgroups of dyslexic readers, where the mean response for reading was used as a dependent variable. In the L-type subgroup, the regression model with text reading accuracy as a predictor was significant (F = 16.87, *p* = 0.001), and accounted for 58.4% of variance. In the P-type group, the regression analyses with Global reading speed and right intermanual tactile errors as predictor scores was computed. The model with Global reading speed was significant (F = 11.91, *p* = 0.001), and accounted for 22.1% of variance. Lastly, in the M-type group, a regression model with Global reading speed and phonemic blending as predictors was significant (F = 8.12, *p* = 0.001), and accounted for 16.2% of total variance (10% Global reading speed, 6.2% phonemic blending) (Figure 3).

#### 3.2.5. Correlations and Regression Models in M-Type Children with Extensive Assessment

Pearson’s correlation analysis was computed in the subgroup of 31 children classified as M-type dyslexic readers. Significant association of mean response for reading were found with Global reading speed and accuracy (r ≤ −0.658, *p* ≤ 0.006), Global writing accuracy (r = −0.327, *p* = 0.072), Global memory score (r = −0.399, *p* = 0.026), Global RAN speed (r = −0.349, *p* = 0.055), and Global visual search accuracy (r = 0.398, *p* = 0.027). Moreover, it was associated with the subcomponents of text and word reading accuracy (r ≤ −0.456, *p* ≤ 0.010), text, word and nonword reading speed (r ≤ −0.526 *p* ≤ 0.002), word writing accuracy (r = −0.414, *p* = 0.021), letter span forward (r = −0.433, *p* = 0.015), RAN-numbers speed (r = −0.341, *p* = 0.061), and Visual search-figures accuracy (r = 0.419, *p* = 0.019). Check for spurious effects with Oldham’s method confirmed validity for the correlation with Global reading speed and accuracy (r ≤ −0.399, *p* ≤ 0.026), and the subcomponents of text and word reading accuracy, and text and word reading speed.

Based on previous correlation analyses, the regression model was computed with mean responses for reading as a dependent variable and Global reading speed and accuracy, letter span forward, Global RAN speed and Visual search-figure accuracy as predictors. A regression model with Global reading speed and Visual search-figure accuracy as predictors was found to be significant (F = 15.48, *p* < 0.001) and account for 52.2% of variance (43.3% Global reading speed, 9.2% Visual search-figure accuracy) (Figure 4).

### 3.3. Definition and Comparison of Responders and Non-Responders for Reading and Writing

According to the mean response in reading and writing, 122 children were classified as responders while the remaining 23 (15.9%) were labelled as non-responders. No significant differences between the two groups emerged in IQ and age (all *p*s > 0.294). In pre-test global scores, responders and non-responders significantly differed only with respect to Global intermanual tactile transfer errors (*p* = 0.048, partial eta square = 0.027). Analysis of the subcomponents showed no significant differences. Pre-treatment data and group comparisons are presented in Table 6.

### 3.4. Associations between Reading and Writing Improvements and Pre-Test Measures

Pearson’s correlation analyses in the large group showed no significant associations of mean response for reading and writing and Global scores (r ≥ −0.126, *p* ≥ 0.132), with the exception of a close-to-significant correlation with Global intermanual tactile errors (r = −0.139, *p* = 0.095). Considering subcomponents, a significant association emerged with the subcomponents of word (r = −0.212, *p* = 0.013) and of nonword writing accuracy (r = −0.250, *p* = 0.003), and left intermanual tactile errors (r = −0.145, *p* = 0.082). Checking for spurious effects with Oldham’s method revealed possible inflation of correlations with writing accuracy due to regression to the mean (r ≤ 0.056, *p* ≥ 0.519).

#### 3.4.1. Regression Analyses

Based on correlation analyses, a regression analysis was conducted with the mean response for reading and writing as a dependent variable in the whole group, and left intermanual tactile errors as predictor. The predictor did not enter the regression model.

Further correlation and regression analyses were then computed in the group of children with all measures of interest including RAN and Visual search, as well as in the subgroups of children previously identified (a. male and female, b. children classified as P-, L-, or M-type dyslexic readers, c. M-type group belonging to the subgroup of 48 children).

#### 3.4.2. Correlations and Regression Models in Children with Extended Assessment (n = 48)

In the subgroup of 48 participants, mean response for reading and writing was significantly associated (negatively) with Global reading speed and accuracy, Global writing accuracy, Global RAN speed and (positively) with Global visual search accuracy. Considering subcomponents of global scores, mean response for reading and writing was significantly associated with text and word reading accuracy (r ≤ −0.374, *p* ≤ 0.009), text and word reading speed (r ≤ −0.254, *p* ≤ 0.082), word, nonword and sentence writing accuracy (r ≤ −0.276, *p* ≤ 0.058), phonemic blending errors (r = 0.324, *p* = 0.025), letter span forward (r = −0.282, *p* = 0.052), and RAN (colors, figures and digits) speed (r ≤ −0.296, *p* ≤ 0.041), visual search-figure speed (r = −0.254, *p* = 0.081) and accuracy (r = 0.278, *p* = 0.055). Pearson’s correlations of global scores are presented in Table 7.

Following checks with Oldham’s method, the correlation with Global reading accuracy was confirmed as valid and not due to regression to the mean (r = −0.317, *p* = 0.028), whereas the correlations with Global reading speed and Global writing accuracy and components of writing accuracy (r ≥ −0.258, *p* ≥ 0.076) did not pass the test and were excluded from regression models. Among subcomponents, correlation with text reading speed, however, was confirmed as valid (r = −0.370, *p* = 0.010) in contrast to other speed scores (r ≥ −0.248, *p* ≥ 0.089) and it was considered in the regression analyses.

A regression model was computed with the mean response for reading and writing as a dependent variable and Global reading accuracy, text reading speed, phonemic blending, letter span forward, Global RAN speed, Global visual search accuracy and visual search-figure speed as predictors. Results showed that the model with text reading speed and RAN speed as predictors was significant (F = 8.10, *p* = 0.001), and accounted for 26.5% of variance (19.5% text reading speed, 7% RAN speed) (Figure 5).

#### 3.4.3. Correlations and Regression Models in Male and Female Children

In the male group, no significant associations were found between Global measures and mean response for reading and writing (r ≥−0.154, *p* ≥ 0.142). In the female group, associations were found with Global writing accuracy (r = −0.454, *p* = 0.001), in all its components of word (r = −0.570, *p* < 0.001), nonword (r = −0.490, *p* < 0.001) and sentence (r = −0.378, *p* = 0.005) writing accuracy, with word reading speed (r = −0.272, *p* = 0.049) and with letter span forward (r = −0.301, *p* = 0.029). A check for spurious effects with Oldham’s method confirmed the validity of all writing scores (r ≤ −0.351, *p* ≤ 0.012) except for sentence writing (r = −0.190, *p* = 0.173), and for word reading speed score (r = −0.146, *p* = 0.296). Regression analysis was computed with word and nonword writing accuracy and letter span forward as predictors. The model with word and nonword writing accuracy as predictors was found to be significant (F = 17.42, *p* < 0.001), and accounted for 42.1% of variance (32.5% word writing, 9.5% nonword writing) (Figure 6).

#### 3.4.4. Correlations and Regression Models in Different Subtypes of Dyslexia

Pearson’s correlation analyses of Global scores (two tailed) on subgroups of children classified according to Bakker’s Balance Model are reported in Table 8.

In L-type dyslexic readers, mean response for reading and writing was (negatively) associated with Global reading accuracy (word reading accuracy, r = −0.619, *p* = 0.018), and Global writing accuracy (word and nonword writing accuracy, r ≥ −0.516, *p* ≤ 0.059). In the P-type group, significant correlations emerged with Global writing accuracy (word, nonword and sentence writing, r ≤ −0.381, *p* ≤ 0.013), word reading accuracy (r = −0.444, *p* = 0.003), and close-to-significant correlations were found with Global intermanual tactile errors (right intermanual tactile score, r = −0.315, *p* = 0.037), and phonemic blending (r = 0.268, *p* = 0.078). Lastly, in the M-type group, a close-to-significant correlation emerged with phonemic blending (r = −0.205, *p* = 0.056). When checking for spurious effects with Oldham’s method in the L-type group, Global reading and writing accuracy and subcomponents did not pass the test (r ≥ −0.519, *p* ≥ 0.057). In the P-type group, Global writing accuracy (and subcomponents of word and sentence) and word reading accuracy did not pass Oldham’s test (r ≥ −0.289, *p* ≥ 0.057), while the correlation with nonword writing accuracy was confirmed as valid (r = −0.484, *p* = 0.001) and it was considered in the regression analyses. Regression analyses were computed in the P-type group of children, where the mean response for reading and writing was used as a dependent variable, and nonword writing accuracy, Global intermanual tactile errors and phonemic blending were used as predictors. Only nonword writing accuracy was found to be a significant predictor (F = 20.41, *p* < 0.001) for P-types, accounting for 34.4% of variance. In the M-type group, phonemic blending did not enter the regression model.

#### 3.4.5. Correlation and Regression Models in M-Type Children with Extended Assessment

Correlation analysis was computed in the subgroup of 31 M-type dyslexic readers who had undergone extended assessment. Significant association of mean response for reading and writing were found with Global reading speed and accuracy (r ≤ −0.434, *p* ≤ 0.015), Global writing accuracy (r = −0.387, *p* = 0.031), Global memory score (r = −0.373, *p* = 0.039), Global RAN speed (r = −0.308, *p* = 0.092), Global visual search accuracy (r = 0.331, *p* = 0.069). Moreover, it was associated with the subcomponents of text, word and nonword reading accuracy (r ≤ −0.379, *p* ≤ 0.035), text and word reading speed (r ≤ −0.386 *p* ≤ 0.032), word and nonword writing accuracy (r ≤ −0.305, *p* ≤ 0.096), letter span forward (r = −0.327, *p* = 0.073), and Visual search-figures accuracy (r = 0.350, *p* = 0.053) and speed (r = −0.314, *p* = 0.085). A check for spurious effects with Oldham’s method confirmed validity for the correlation with Global reading accuracy (r = −3.96, *p* = 0.028) and the subcomponents of nonword reading accuracy and text reading speed, whereas correlations with word and text reading accuracy only approached significance.

Based on previous correlation results, a regression model was computed with the mean response for reading and writing as a dependent variable and Global reading accuracy, text reading speed, Global memory score, Global RAN speed, Visual search-figure speed and accuracy as predictors. Only text reading speed turned out to be a significant predictor (F = 8.57, *p* = 0.007) accounting for 22.8% of variance.

## 4. Discussion

### 4.1. Characteristics of Non-Responders

The first aim of the present retrospective study was to explore the specific markers of non-responders to treatment as well as the predictors of improvement in reading and spelling skills after a neuropsychological treatment based on VHSS, in children with a diagnosis of DD. Response to intervention was analyzed, both taking into account only improvement in reading and taking into account the average of improvements in reading and writing. The comparison of the two groups of children subdivided into responders and non-responders according to their improvement in reading only (response for reading) showed two distinctive patterns of performance at pre-test and an Intervention Differential Effect (IDE) [65]. Group differences emerged for writing accuracy—more specifically, for nonword writing accuracy, followed by sentence writing accuracy—and for nonword reading accuracy. Moreover, a close-to-significant difference was found for errors in phonemic awareness tasks. Thus, children who do not improve in reading are those starting with the lowest scores in writing, especially following the indirect, sublexical route (needed for writing nonwords) and with the highest number of errors in phonemic blending and elision, but they are less impaired in nonword reading accuracy (also requiring the use of the indirect route, but in a different modality as compared to writing, having to segment nonwords in their visual, as opposed to their auditory, form). This pattern of results is actually consistent: non-responders are characterized by a difficulty in segmenting nonwords at the auditory level in order to write them, and this difficulty is confirmed by the high number of errors in phonemic awareness tasks. By contrast, they have less difficulties in using the segmental route starting from the visual modality in order to read nonwords: indeed, they are less impaired than responders in this task. We may thus hypothesize (even if no direct measure of auditory processing was included in the present data) that non-responders for reading have specific difficulties in processing phonological representations, but not in visual processing.

Turning to the response to intervention with respect to both reading and writing, the performance pattern at pre-test characteristic of non-responders changes: in this case, the only difference concerns intermanual errors at the task of tactile callosal transfer. Precisely, non-responders commit more errors in transferring of tactile information (with excluded sight) to the opposite hand, regardless of side. Since they do not commit a comparable number of errors in the intramanual condition (not needing information to be transferred via the corpus callosum), this suggests that their problem does not lie in the identification of the finger that has been touched, but in the transfer to the other hand for responding. In other terms, they have impaired or inefficient interhemispheric communication.

The role of the corpus callosum in response to the treatment of reading and writing difficulties has seldom been specifically investigated in the experimental literature. Evidence concerning its involvement in writing improvement comes from previous studies by our research group [26,33]. Nonetheless, the role of inter-hemispheric communication in reading has been more widely studied and interesting reports exist on the presence of anomalous performances in callosal-based tasks in individuals with developmental dyslexia [66,67,68,69,70,71], also supported by (rather contradictory) data from neuroimaging studies (e.g., [72,73]). Indeed, the idea of optimal inter-hemispheric communication is at the very basis of the VHSS treatment and of the Balance Model, so the fact that children who have more severe impairments in transferring information from one hemisphere to the other one are also less likely to profit from intervention is compatible with the model, assuming that a severe impairment could be more difficult to overcome due to structural reasons. This is actually different from the general principles of the Balance Model, positing that neurocognitive organization can be modified through training and that flexibility is the rule, but it is also true that the brain’s general functional organization could be more flexible than structurally constrained functions in the corpus callosum (differences in neuroplasticity are reported for grey and white matter, and callosal structures seem to be responsive to environmental effects during restricted sensitive periods) [74,75,76].

### 4.2. Associations between Response to Intervention and Pre-Test Performance

A second aim of the study was to identify the associations of measures of response to intervention and pre-test scores. This differs from the first aim, because the characteristics of non-responders may concern only a small subgroup of children who are “resistant to treatment” but do not necessarily describe the general patterns of association emerging in the whole group.

Results on the associations between mean response in reading and all the pre-test measures showed that children who have had the greatest improvement in reading were more impaired in initial values of reading accuracy and speed, but less impaired in intermanual tactile information transfer. While negative association with initial accuracy in reading could have been inflated by spurious effects due to regression to the mean, the other associations were robust and reliable. In other terms, improvement was negatively associated with initial reading speed and positively with callosal functions. Thus, the role of inter-hemisphere communication is not limited to the small group of children who did not respond to treatment, but it is extended to the whole group of children with dyslexia, showing that the ability to optimally establish connections between the two hemispheres is a crucial factor in intervention efficacy. This is in line with previous studies highlighting the importance of callosal connections in reading and dyslexia [66,67,68,69,70,71,72,73], and with hypotheses addressing the role of callosal functions with respect to reading improvement over time [77,78]. Indeed, the two hemispheres have been described [79,80] as competing in development by mutual inhibition controlled by the corpus callosum, which would entail a dysfunctional corpus callosum not only possibly limiting active maturation of the left reading network, but also leading to hyperdevelopment of homologous regions of the right hemisphere. No associations with age and IQ emerged in our data.

In the subgroup of 48 children who had undergone more extensive assessment, mean response for reading was significantly and negatively associated with reading speed and accuracy, writing accuracy (especially for words), phonemic awareness (especially phonemic blending), RAN speed, and letter span forward. This confirms what had been shown by previous studies [10,41,42,43,46].

Further analyses in distinct groups of males and females revealed, for males, reliable correlations with reading speed. In the female group, no associations survived Oldham’s check for inflation through regression to the mean. As for dyslexia subtypes, L-types showed reliable associations with text reading accuracy, P-types with reading speed, and M-types with reading speed and with phonemic blending. These patterns may be seen as mirroring the typical weaknesses of each subtype [23,24].

Results on the associations between the mean response in reading and writing and all the pre-test measures showed that children who had the greatest improvement in reading/writing were less impaired in initial values of visual search accuracy, and more impaired in reading speed and accuracy, writing accuracy and RAN speed. Females showed reliable associations with writing accuracy and letter span forward. No associations were found for males. As for subtypes, L-type dyslexic readers showed no reliable associations, P-types showed associations with nonword writing accuracy and right intermanual tactile scores, and no significant associations emerged for M-types. The interpretation of these correlations will be further addressed in the next section, after analysis of their actual involvement in predictive models.

### 4.3. Predictive Models

The third and last aim of the study was to identify predictors of reading and writing improvement through linear multiple regression models. This adds information to simple correlations because it allows us to investigate the interplay between different functions that are all involved in improvement, but whose contribution may overlap and interact with the contribution of other variables. Starting with predictors of reading improvement, regression models identified the contribution of global reading speed, accounting for 14.3% of total variance in the whole group. Measures of inter-hemispheric communication, by contrast, did not enter regression models as a significant predictor of improvement, and the same was true for measures of verbal memory.

When children with additional neuropsychological measures were considered (the group of 48 children), only reading accuracy was added to the model. Analyzing these results with respect to subgroups for gender and dyslexia subtypes, it turned out that reading speed had a significant predicting value for females (in association with nonword writing accuracy) but not for males. More interestingly, it was found that reading accuracy (especially for text) was predictive for L-types, whereas reading speed was predictive for P-types (in both cases, the predictor coincides with the specific weakness of the subtype, i.e., with the parameter which is likely to be represented by the lowest scores, and thus allowing for more improvement). In M-types, blending entered the model in addition to reading speed. In all cases, reading speed accounted for a larger proportion of variance compared to other predictors. The only exception was L-types, for whom reading accuracy (the specific weakness of the subtype) turned out to be a very strong predictor. In M-types with extended assessment, reading speed and visual search accuracy accounted for over 50% of variance (precisely, 43.3% reading speed and 9.2% visual search).

As for improvement in reading and writing, a regression model with text reading speed and RAN speed as predictors explained 26.5% of variance in the sample of 48 children with extended assessment (19.5% text reading speed, 7% RAN speed). For females only, a model with word and nonword writing accuracy as predictors accounted for over 42% of variance. Finally, nonword writing accuracy in P-types accounted for 34.4% of variance. In M-types with extended assessment, text reading speed turned out to be a significant predictor accounting for 22.8% of variance.

It can thus be concluded that initial reading speed is a better predictor of response to intervention than reading accuracy. This does not mean, however, that reading speed improves more than reading accuracy: indeed, the degree of improvement depends on the initial profile of reading strengths and weaknesses (in the whole sample, accuracy improved an average of 1.02 z-scores and speed of 0.75 z-scores, but accuracy improvement was 1.37 in L-types, 0.93 in P-types and 1.02 in M-types; and speed improvement was 0.34 in L-types, 1.07 in P-types and 0.65 in M-types, exactly reflecting the weaknesses of each subtype). Rather, it points to the more reliable and more stable nature of reading speed in dyslexia: in fact, associations of reading accuracy with improvement were found more than once to (at least partially) depend on regression to the mean, whereas this did not happen for reading speed. Indeed, while reading errors are sporadic events that can be more subjected to random variations, reading speed is a continuous variable characterizing performance as a more fine-grained and stable feature of reading. The more revealing and significant role of reading speed in dyslexia for populations speaking orthographically transparent languages has been highlighted in the literature [81].

The role of blending as a predictor of reading improvement in M-types is in line with previous studies pointing to phonemic awareness as a major predictor of response to intervention in dyslexia [10,16,82].

Also, the predictive role of RAN was described in previous literature. Interestingly, other studies [10] found that RAN speed was the only predictor for all types of improvement under study, involving the direct and indirect route of reading at different degrees. Using mediation models, Tilanus and co-authors [16] found a direct effect of pre-test RAN measures on children’s progress in speed of grapheme-to-phoneme conversion, as well as an indirect effect on word and pseudoword reading; the same group [17] found that RAN ability at pre-test directly predicted post-test scores for word and pseudoword reading, and explained it as an expression of the strict correlation between reading fluency and RAN proficiency [47,83].

More surprising is the predictive effect of writing measures with respect to reading improvement. Indeed, orthographic skills were found to predict reading progress [84]. It can be observed that the role of reading speed as a predictor for reading improvement is taken over by writing (especially nonword writing on dictation) in the prediction of reading and writing improvement. This suggests that the improvement in writing can occur only if all neuropsychological routes to writing are activated (a direct route accessing whole orthographic representations and an indirect route passing through phoneme-to-grapheme conversion) [85].

It should be noted that the analysis applied in regression models for the present study substantially differs from most previously published papers on the topic, aiming to predict post-test performance based on pre-test scores (e.g., [16,17,18,84]). In the case of previous studies, pre-test reading scores expectedly (and positively) predicted post-test reading scores (for this reason, they were set as control variables or directly partialled out in some of the analyses). In the present study, by contrast, what is predicted is the degree of improvement irrespective of the final level of performance. Therefore, the predictive power of pre-test reading scores is not a completely granted result (although children with lower initial scores naturally have larger space for improvement and children not impaired at the beginning are less likely to show large variation in response to treatment), and its central role surpassing the effects of any other pre-test variable was not necessarily expected. Indeed, in clinical contexts it is sometimes even assumed that more severe cases of reading and writing impairments could be more resistant to intervention and show the least improvement. Our results show that this is not the case, and that by contrast lower initial levels predict larger improvements. The only confounding effect could be due to regression to the mean (the statistical phenomenon according to which the most extreme cases of random variability tend to be less extreme in a repeated measurement, implying that performances that were extremely low at pre-test would tend to improve at post-test for mere statistical reasons). When pre-test variables are also used as a cut-off for inclusion (this is the case for inclusion criteria applied to reading scores for children with dyslexia) the result is that regression to the mean in the negative direction (i.e., very low scores due to random variation) is not counterbalanced by regression to the mean in the positive direction (very high pre-test scores due to random variation would result in exclusion of the participant from the sample). For this reason, all correlations between improvement and pre-test reading and spelling scores (those that were also used as inclusion criteria) were checked for the presence of regression to the mean effects via Oldham’s method (controlling that the same associations were present when averaging pre-test and post-test scores). Only variables whose associations with improvement passed this test were included in regression analyses as predictors, ensuring that the resulting model was not reflecting spurious effects of random variation.

Both mathematical procedures (the one predicting post-test scores and the one predicting change scores) have inherent limitations but also strengths. They actually reflect two slightly different perspectives, which may address different issues and goals. The first perspective, focusing on the end-point reached after intervention, may be especially relevant for its impact on school proficiency and progress. The second perspective, focusing on the degree of improvement, may be considered to be a more relevant perspective for clinical researchers, who need to judge the efficacy and efficiency of intervention procedures, and may consider starting and final performance levels in order to activate repeated cycles or additional support strategies. In this perspective, the highly predictive power of pre-test reading scores, which turned out to be larger than the predictive power of phonological awareness, phonological memory and visual attention, should be regarded as an interesting and encouraging result. According to our data, the only neuropsychological characteristics that might pose limits to improvement are poor visual attention and poor interhemispheric communication, while poor phonological awareness (both phoneme blending and RAN) leaves space for improvement even in severe cases.

### 4.4. Subgroup Specificities

Due to the rather low proportion of total variance that was explained by the predictors in regression models, further analyses were conducted to investigate whether different predictive patterns could characterize different subgroups of dyslexic children. These further analyses were conducted starting only from the larger group of 145 children, in order to not obtain too-small groups. This implies that the two variables RAN and visual search, that were available only for 48 participants, were excluded (with the exception of M-types, who were numerous enough to allow for separate analyses considering all variables).

Two kinds of distinctions were taken into account: gender differences and subtypes of dyslexia. In both cases, it can be observed that focusing on more homogeneous subgroups allowed explanation of a definitely larger amount of variance, up to half of it.

The first distinction, according to gender, showed that—for females only—reading speed predicts improvement of reading, while word and nonword writing accuracy predict improvement in reading and writing. The greater difficulty in predicting response in males is consistent with previous studies indicating that male persons with dyslexia show larger variance in reading measures, slower and more variable processing speed and worse inhibitory control [86,87]. Moreover, writing accuracy, which showed to be a weaker predictor compared to reading speed, does play a relevant role in female individuals with dyslexia. This might be due to orthography-based coding and decoding strategies possibly being preferentially adopted by females, who are known to have a general advantage in language and in orthography measures compared to males [88,89,90].

The second distinction, based on dyslexia subtypes, showed results generally consistent with what is already known about subtyping according to the prevalence of speed vs. accuracy impairments in reading. In fact, subtypes tend to improve, especially in the variables that represent a specific weakness (speed for P-types, accuracy in L-types, both speed and accuracy in M-types). Indeed, the treatment based on VHSS provides a stimulation of all reading-related processes, but especially targets strategies aimed at overcoming subtype-specific weaknesses (accurate decoding in L-types, whole-word recognition and language-based anticipation for P-types, both strategies in M-types) [24,26]. In the additional analysis conducted on the whole set of variables for M-types, reading speed and visual search accuracy were found to predict improvement in reading speed (the first variable as a negative predictor, the latter as a positive predictor). Greater impairments in visual attention point to worse conditions for intervention effectiveness: children who are more chaotic in their visual search strategies seem to profit less from training. Since visual-spatial attention as measured by Posner paradigms and cue effect size has been shown in previous studies [91] to greatly improve following VHSS treatment, it is not plausible to hypothesize that it would not be the case for the present children. Therefore, a more plausible hypothesis is that the function measured by visual search and whose impairment would be hampering improvement is more related to frontal-based executive functions and top-down processes [92] than to parietal-controlled visual-spatial attention. Indeed, pre-school visual search measures turned out to be very good predictors of first-grade reading in previous studies [21,93]. The relationship of visual attention with response to intervention has been much less investigated, but it might be a relevant factor in improvement.

All other variables (such as memory and callosal transfer) that turned out to be correlated to improvement did not finally enter regression models as predictive variables.

### 4.5. Limitations and Future Perspectives

The present study has a series of limitations. First of all, some of the variables of interest are available for only part of the sample (48 children), making it impossible to apply further distinctions (such as gender and dyslexia subtype, except for M-types) to the whole set of variables. The only exception is M-types, for whom reading speed and visual search accuracy were, in fact, found to predict improvement in reading speed. It remains unclear what role lexical access (as measured by RAN) and visual attention (as measured by visual search) could have on the remaining subgroups (P-types and L-types, or males and females separately). This is clearly a primary aim to be pursued in future studies. Additionally, the lack of information on SES in the present sample could obscure some relevant effects on responses to intervention, as a recent study has found that SES is negatively associated with reading improvement (and cortical thickening) after intensive intervention [94].

A further limitation is the possibility that parts of the findings are influenced by mathematical constraints (even if major artefacts were excluded through rigorous checks and by removing from regression models all variables that might have been flawed by regression to the mean), due to larger space for improvement being intrinsically granted for scores that are lower at pre-test. This may explain some of the negative correlations emerging with all pre-test scores. However, since correlations with reading/writing improvement extend to non-reading variables such as memory, phonemic awareness and RAN, whose changes were not measured as dependent variables, and since positive instead of negative correlations were found with other variables such as callosal transfer (i.e., a negative correlation with the number of errors) and accuracy of visual search, mathematical artefacts clearly cannot be the main source of observed associations.

A final weakness relates to the arbitrary cut-off used for the definition and identification of non-responders. This choice has already been discussed in the Data Analysis section, where some data are reported supporting its plausibility. Indeed, large variability in the definition and criteria for identifying non-responsiveness to treatment have been reported in previous review studies [10] and the percentages of non-responders identified in the present study (12% to 16%) are in line with average percentages previously reported.

It should also be highlighted that the generalizability of the present results is constrained by the specific type of intervention that was employed in the study, and by the language and orthographic system (Italian has a very transparent orthography compared to other Western languages and its orthographic structure is even more different from most Eastern written language systems, sometimes characterized by the use of ideograms and/or multiple coding systems as in Chinese and Japanese, or by diglossia as in several Arabic languages). Indeed, VHSS is not very widely applied for the treatment of dyslexia, although it has shown good efficiency and efficacy [24,26,28,33]. This may in part be due to the fact that the theoretical model on which the program is based is rather old [23] and certainly cannot be considered as an up-to-date model of the neural circuitries of reading and dyslexia. Nonetheless, as highlighted in the introduction, recent findings form neuroimaging studies [30,31,32] have provided evidence compatible with the model, and new contributions [77,78,79,80] provide interesting interpretations that give new light to aspects of the model that were under-specified or obscure in its initial formulations. We believe that the present findings can also contribute to reframing the model in a contemporary framework where the role of callosal connections and hemisphere cooperation are substantially preserved and confirm the main principles of the intervention program, possibly shifting the focus on the crucial effects of timing and synchronicity in maturational processes affecting different brain structures [77]. Clearly, replication of these results by other independent groups would be desirable to increase generalizability and/or highlight language- or context-specific constraints on the activated mechanisms of improvement.

## 5. Conclusions

Summing up the many results of the study, it can be concluded that reading improvement in response to neuropsychological treatment following VHSS intervention mainly depends on the level of reading impairments at pre-test (with greater impairments leading to greater improvements). A minor role is played by phonological awareness (phoneme blending) and visual search accuracy, whose role emerged only for M-type dyslexic readers, thus excluding children who have more specific decoding strategies as P-types and L-types. Improvement in reading and writing, by contrast, depends on initial levels of writing impairment, especially writing through indirect, phoneme-to-grapheme conversion (again, in a negative relationship implying that phoneme-to-grapheme conversion is not a strategy that is already present at pre-test, but rather one that needs to be developed through treatment). Additional factors supporting improvement like RAN speed, phonemic blending and visual search play a relevant (even if minor, never exceeding 10% explained variance) role.

Phoneme blending for reading and RAN speed and nonword writing accuracy for reading and writing may be seen as the bases for the activation of both direct and indirect strategies to reading (blending supporting phoneme assembly after grapheme-to-phoneme conversion) and writing (RAN pointing to the retrieval of whole word representation, whereas nonword writing proceeds in a segmental way).

Memory skills and interhemispheric transfer showed correlations with improvement but did not significantly participate to the explanation of its variance (possibly as a consequence of the conservative cut-offs for regression not allowing them to enter the models). Age and IQ were not significantly associated with improvement.

The present results support the hypothesis that characteristics of non-responders do not necessarily relate to the same variables predicting improvement in the whole group. Specific markers that can point to an enhanced risk that a child will not respond positively to treatment include very low writing scores at pre-test, encompassing both the direct and the indirect route to writing; when writing is also the target of intervention, an additional marker of non-responders is a severe impairment in callosal transfer of tactile information, suggesting limited support from interhemispheric integration to the writing process.

## Figures and Tables

**Figure 1 brainsci-14-00775-f001:**
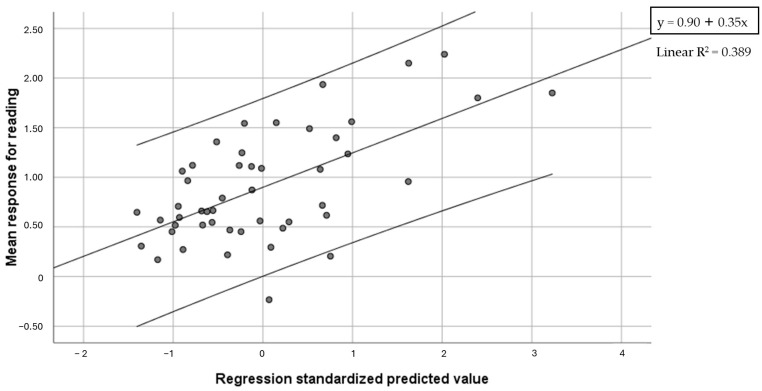
Regression plot with mean response for reading as a dependent variable and Global reading speed and Global reading accuracy as predictors. Confidence intervals (95%) are represented. Regression equation in the upper right rectangle.

**Figure 2 brainsci-14-00775-f002:**
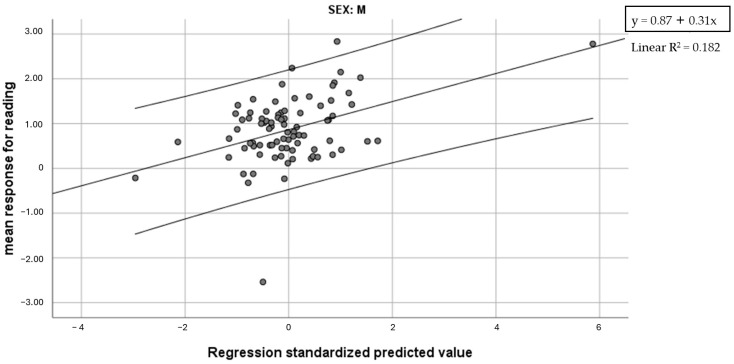
Regression plot with mean response for reading as a dependent variable and Global reading speed and nonword writing accuracy as predictors. Confidence intervals (95%) are represented. Regression equation in the upper right rectangle.

**Figure 3 brainsci-14-00775-f003:**
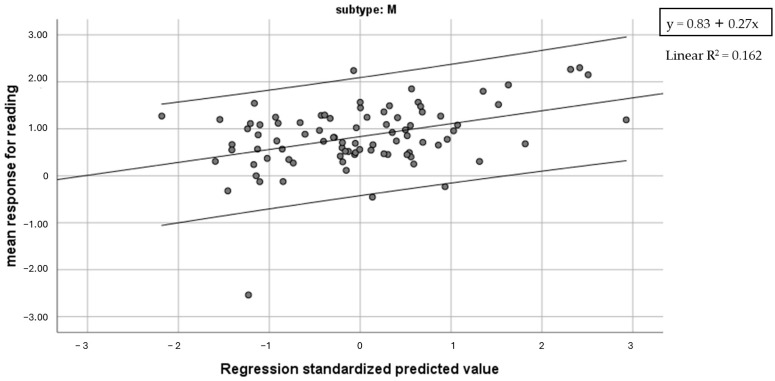
Regression plot with mean response for reading as dependent variable and Global reading speed and phonemic blending in the M-type group of dyslexic readers. Confidence intervals (95%) are represented. Regression equation in the upper right rectangle.

**Figure 4 brainsci-14-00775-f004:**
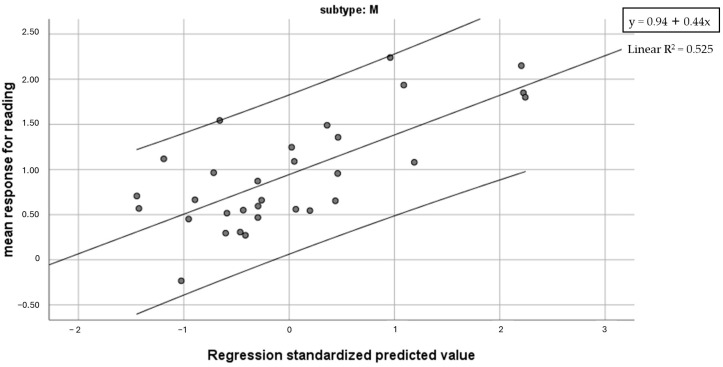
Regression plot with mean response for reading as dependent variable and Global reading speed and visual search (figures) accuracy in M-types with extended assessment. Confidence intervals (95%) are represented. Regression equation in the upper right rectangle.

**Figure 5 brainsci-14-00775-f005:**
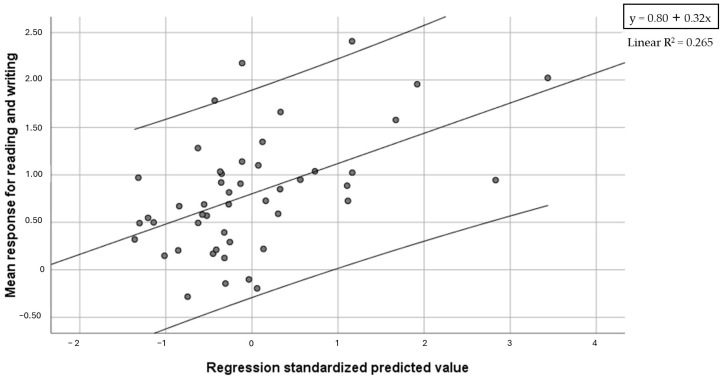
Regression plot with the mean response for reading and writing as a dependent variable and text reading speed and RAN speed as predictors in the group of 48 children with extended assessment. Confidence intervals (95%) are represented. Regression equation in the upper right rectangle.

**Figure 6 brainsci-14-00775-f006:**
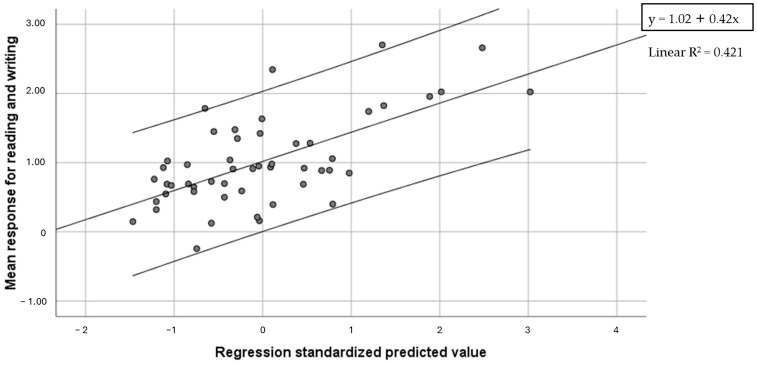
Regression plot for female participants, with the mean response for reading and writing as a dependent variable and word and nonword writing accuracy as predictors. Confidence intervals (95%) are represented. Regression equation in the upper right rectangle.

**Table 1 brainsci-14-00775-t001:** Means and SDs of pre- and post-test global scores.

	Pre-Test Mean (*SD*)	Post-Test Mean (*SD*)
IQ	99.69 (12.84)	/
Global reading accuracy (z-score)	−2.37 (1.55)	−1.35 (1.49)
Global reading speed (z-score)	−2.32 (2.20)	−1.57 (2.04)
Global writing accuracy (z-score)	−2.49 (2.79)	−1.62 (3.12)
Global phonemic awareness errors	4.29 (2.88)	2.72 (2.59)
Global memory score (z-score)	−0.48 (0.55)	−0.06 (0.67)
Intramanual left hand errors	0.46 (0.79)	0.44 (0.85)
Intramanual right hand errors	0.53 (1.03)	0.36 (0.80)
Intermanual left hand errors	5.03 (3.96)	4.04 (3.65)
Intermanual right hand errors	4.74 (3.88)	3.94 (3.63)

**Table 2 brainsci-14-00775-t002:** Pre-test Means and SDs of responders’ and non-responders’ scores (considering only reading improvements) on Global variables and results of the ANOVA comparing the two groups. Significant differences are shown in bold.

	Responders n = 128	Non-responders n = 17	Pre-Test between-Group Comparison
	Mean (*SD*)	Mean (*SD*)	MANOVA F, *p* (Two-Tailed), Partial eta Squared
Global reading accuracy (z-score)	−2.45 (1.53)	−1.75 (1.62)	3.08, 0.081, 0.021
Global reading speed (z-score)	−2.37 (2.29)	−1.89 (1.35)	0.71, 0.400, 0.005
Global writing accuracy (z-score)	−2.31 (2.60)	−3.88 (3.70)	**4.88, 0.029, 0.033**
Global phonemic awareness errors	4.13 (2.67)	5.47 (4.06)	3.31, 0.071, 0.023
Global memory score (z-score)	−0.50 (0.55)	−0.36 (0.55)	1.01, 0.317, 0.007
Global intramanual tactile errors	0.50 (0.74)	0.44 (0.68)	0.09, 0.757, 0.001
Global intermanual tactile errors	4.75 (3.62)	5.85 (3.37)	1.40, 0.238, 0.010

**Table 3 brainsci-14-00775-t003:** Pearson’s correlations between mean response for reading and Global scores (N = 145).

	Global Reading Speed	Global Reading Accuracy	Global Writing Accuracy	Global Phonemic Awareness Errors	Global Memory Score	Global Intramanual Tactile Errors	Global Intermanual Tactile Errors
	r, *p*						
**Mean response for reading**	−0.379, <0.001	−0.271, 0.001	0.055, 0.511	−0.118, 0.159	−0.128, 0.124	−0.135, 0.105	−0.141, 0.090

**Table 4 brainsci-14-00775-t004:** Pearson’s correlations between mean response for reading and Global scores (N = 48). (Intram. = Intramanual; Interm. = Intermanual).

	Global Reading Speed	Global Reading Accuracy	Global Writing Accuracy	Global Phoneme Awarareness	Global Memory Score	Global Intram. Tactile Errors	Global Interm. Tactile Errors	Global RAN Speed	Global RAN Accuracy	Global Visual Search Speed	Global Visual Search Accuracy
r, *p*											
**Mean response for reading**	−0.521, <0.001	−0.488, <0.001	−0.298, 0.040	0.221, 0.132	−0.296, 0.041	−0.157, 0.286	−0.249, 0.087	−0.286, 0.048	0.037, 0.803	−0.144, 0.441	0.272, 0.062

**Table 5 brainsci-14-00775-t005:** Pearson’s correlations between mean response for reading and Global scores in the three subgroups of dyslexic readers (N = 145).

		Global Reading Speed	Global Reading Accuracy	Global Writing Accuracy	Global Phonemic Awareness Errors	Global Memory Score	Global Intramanual Tactile Errors	Global Intermanual Tactile Errors
r, *p*								
**Mean response for reading**	L	−0.380, 0.180	−0.683, 0.007	0.028, 0.925	−0.286, 0.321	0.375, 0.187	0.190, 0.515	−0.251, 0.386
	P	−0.470, 0.001	−0.237, 0.122	0.092, 0.553	0.072, 0.642	−0.193, 0.209	−0.176, 0.254	−0.214, 0.163
	M	−0.316, 0.003	−0.265, 0.013	0.029, 0.790	−0.174, 0.108	−0.153, 0.157	−0.127, 0.242	−0.081, 0.454

**Table 6 brainsci-14-00775-t006:** Means and SDs of responders’ and non-responders’ scores (considering both reading and writing improvements) on Global variables and results of the ANOVA comparing the two groups. Significant differences are shown in bold.

	Responders n = 122	Non-responders n = 23	Pre-Test between-Group Comparison
	Mean (*SD*)	Mean (*SD*)	ANOVA F, *p* (Two-Tailed), Partial eta Squared
Global reading accuracy (z-score)	−2.42 (1.44)	−2.11(2.08)	0.76, 0.385, 0.005
Global reading speed (z-score)	−2.36 (2.29)	−2.08(1.71)	0.30, 0.582, 0.002
Global writing accuracy (z-score)	−2.46 (2.59)	−2.69 (3.74)	0.13, 0.716, 0.001
Global phonemic awareness errors	4.12 (2.74)	5.15 (3.47)	2.49, 0.116, 0.017
Global memory score (z-score)	−0.49 (0.55)	−0.45 (0.57)	0.08, 0.785, 0.001
Global intramanual tactile errors	0.46 (0.59)	0.67 (1.25)	1.67, 0.199, 0.012
Global intermanual tactile errors	4.63 (3.50)	6.24 (3.87)	**3.97, 0.048, 0.027**

**Table 7 brainsci-14-00775-t007:** Pearson’s correlations between mean response for reading and writing and global scores (n = 48). (Intram. = Intramanual; Interm. = Intermanual).

	Global Reading Speed	Global Reading Accuracy	Global Writing Accuracy	Global Phonemic Awareness Errors	Global Memory Score	Global Intram. Tactile Errors	Global Interm. Tactile Errors	Global RAN Speed	Global RAN Accuracy	Global Visual Search Speed	Global Visual Search Accuracy
r, *p*											
**Mean response for reading and writing**	−0.403, 0.004	−0.405, 0.004	−0.403, 0.005	0.268, 0.066	−0.261, 0.073	−0.100, 0.500	−0.213, 0.146	−0.393, 0.006	−0.024, 0.874	−0.152, 0.303	0.323, 0.025

**Table 8 brainsci-14-00775-t008:** Pearson’s correlations between mean response for reading and writing and Global scores in the three subgroups of dyslexic readers.

		Global Reading Speed	Global Reading Accuracy	Global Writing Accuracy	Global Phonemic Awareness Errors	Global Memory Score	Global Intramanual Tactile Errors	Global Intermanual Tactile Errors
r, *p*								
**Mean response for reading and writing**	L	−0.192, 0.511	−0.596, 0.024	−0.617, 0.019	0.244, 0.400	0.008, 0.979	0.139, 0.636	−0.016, 0.958
	P	−0.015, 0.924	−0.242, 0.114	−0.498, 0.001	0191, 0.215	0.046, 0.767	0.071, 0.649	−0.284, 0.062
	M	−0.014, 0.896	0.041, 0.708	0.093, 0.394	−0.145, 0.181	−0.024, 0.824	−0.157, 0.147	−0.085, 0.432

## Data Availability

The data presented in this study are available from the corresponding author, on request and under appropriate sharing agreement, due to restrictions from the Ethical Committee.

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
