# Peer review of "Predicting Response to Neuropsychological Intervention in Developmental Dyslexia: A Retrospective Study"

_brainsci, 2024, doi:10.3390/brainsci14080775_

Round 1

Reviewer 1 Report

Comments and Suggestions for Authors

The aim of this paper is to investigate an important topic in dyslexia research, that of identifying potential non-responders to therapy.

The Visual Hemisphere Specific Stimulation (VHSS) program, which is the focus of the present work, is based on the principle of a tachistoscopic presentation of words written in the contralateral visual hemifield to the under-exploited hemisphere in order to rebalance the involvement of each hemisphere during reading.

It derives from a rather old hypothesis, proposed more than 50 years ago, according to which there are two types of dyslexics, the "P-types" whose decoding strategies are based on an efficient perceptive analysis, mainly supported by the right hemisphere, characterized by a slow but accurate reading, and the "L-types" based on linguistic decoding strategies, involving the left hemisphere, characterized by faster reading but with more errors. It should be noted that this two-type design was proposed before the advent of neuroimaging techniques and therefore does not take into account the considerable progress these techniques yielded in understanding dyslexia.

It is therefore a very specific method and, it should be emphasized, rather ‘confidential’, insofar as it does not appear as such in the most recent meta-analyses and review papers about rehabilitation in dyslexia, and given that the research thereof has almost all been carried out by the same Italian team, precisely that of the present study. It is obvious that this significantly limits its scope as regards any generalization of the results of this study. This should be mentioned in the discussion section.

The point, here, is not that of demonstrating the effectiveness of the said method, which is implicitly considered as granted, but of identifying the factors capable of predicting its effectiveness. To do this, the authors propose a complex statistical analysis that they believe would demonstrate the differential implication of certain predictors, by distinguishing non-responders for read-only from non-responders for reading and writing. Specifically, the improvement in reading is predicted by the score in writing accuracy before training, while the improvement in reading and writing is predicted by the score on an interhemispheric transfer task.

Notwithstanding the above-mentioned limitations, the results are clearly exposed and in themselves convincingly detailed, demonstrating the possibility of predicting the outcome of one type, even confidential, of dyslexia treatment. These results can thus be considered as indirect proof of the usefulness of this therapeutic method and deserve sharing with the scientific community

Comments on the Quality of English Language

OK

Reviewer 2 Report

Comments and Suggestions for Authors

This manuscript focuses on statistical analysis to investigate the relationship between VHSS intervention response and various predictors. This manuscript did a comprehensive linear regression analysis on various variables, including the DD subtypes, participants gender, etc. Here are my comments:

1.     Please specify all statistical tests and significance criteria (e.g., MANOVA, ANOVA) in the methods section.

2.     While the Results section provides detailed statistical analysis, it lacks clear take-home messages for each subsection. Instead, the discussion appears to be a summary of the results, particularly in 4.2. Please improve the Discussion section by linking your findings with those of other studies, offering a more comprehensive interpretation.

Comments on the Quality of English Language

Good.
